# Extension of human lncRNA transcripts by RACE coupled with long-read high-throughput sequencing (RACE-Seq)

Julien Lagarde[1,2,*], Barbara Uszczynska-Ratajczak[1,2,*], Javier Santoyo-Lopez[3,†], Jose Manuel Gonzalez[4], Electra Tapanari[4,†], Jonathan M. Mudge[4], Charles A. Steward[4], Laurens Wilming[4], Andrea Tanzer[1,2,†], Cédric Howald[5,†], Jacqueline Chrast[5], Alicia Vela-Boza[3,6], Antonio Rueda[3], Francisco J. Lopez-Domingo[3], Joaquin Dopazo[3,7,8], Alexandre Reymond[5], Roderic Guigó[1,2] & Jennifer Harrow[4]

Long non-coding RNAs (lncRNAs) constitute a large, yet mostly uncharacterized fraction of the mammalian transcriptome. Such characterization requires a comprehensive, high-quality annotation of their gene structure and boundaries, which is currently lacking. Here we describe RACE-Seq, an experimental workflow designed to address this based on RACE (rapid amplification of cDNA ends) and long-read RNA sequencing. We apply RACE-Seq to 398 human lncRNA genes in seven tissues, leading to the discovery of 2,556 on-target, novel transcripts. About 60% of the targeted loci are extended in either 5′ or 3′, often reaching genomic hallmarks of gene boundaries. Analysis of the novel transcripts suggests that lncRNAs are as long, have as many exons and undergo as much alternative splicing as protein-coding genes, contrary to current assumptions. Overall, we show that RACE-Seq is an effective tool to annotate an organism's deep transcriptome, and compares favourably to other targeted sequencing techniques.

[1] Centre for Genomic Regulation (CRG), Barcelona Institute of Science and Technology (BIST), Dr Aiguader 88, 08003 Barcelona, Spain. [2] Universitat Pompeu Fabra (UPF), Barcelona, Spain. [3] Genomics and Bioinformatics Platform of Andalusia (GBPA), 41092 Seville, Spain. [4] Wellcome Trust Sanger Institute, Hinxton, Cambridgeshire CB10 1HH, UK. [5] Center for Integrative Genomics, University of Lausanne, Lausanne, Switzerland. [6] Roche Diagnostics, 08174 Sant Cugat Del Vallès, Barcelona, Spain. [7] Computational Genomics Department, Centro de Investigación Príncipe Felipe, 46012 Valencia, Spain. [8] Functional Genomics Node (INB), Centro de Investigación Príncipe Felipe, 46012 Valencia, Spain. * These authors contributed equally to this work. † Present addresses: Edinburgh Genomics, The Roslin Institute and R(D)SVS, University of Edinburgh, Easter Bush, Edinburgh EH25 9RG, UK (J.S.-L.); European Molecular Biology Laboratory-European Bioinformatics Institute (EMBL-EBI), Hinxton, Cambridge, UK (E.T.); Department of Theoretical Chemistry, University of Vienna, Waehringerstrasse 17, 1090 Vienna, Austria (A.T.); Division of Genetic Medicine, Geneva University Hospitals, Geneva, Switzerland (C.H.). Correspondence and requests for materials should be addressed to R.G. (email: roderic.guigo@crg.cat) or to J.H. (email: jla1@sanger.ac.uk).

The mammalian transcriptome is composed of a complex mixture of protein-coding and non-protein-coding RNA molecules. Increasing interest has been brought to bear on the latter, most notably on long non-coding RNAs (lncRNAs). A small but growing number of lncRNAs has been reported to play diverse roles in biological and pathological processes[1,2]; however, the vast majority still awaits functional characterization. Such characterization depends on accurate and comprehensive annotation of the complete repertoire of lncRNA transcript structures. This has been the focus of considerable efforts in recent years[3–7]. Arguably the most refined and widely used lncRNA annotation is the catalogue of 15,000 human lncRNA loci published by GENCODE, alongside the Encyclopedia of DNA Elements (ENCODE) data release in 2012 (ref. 8). Various consortia, including the 1,000 genomes project[9], The Cancer Genome Atlas (TCGA)[10] and The Genotype-Tissue Expression (GTEx) Consortium[11] use GENCODE as their reference annotation.

LncRNA gene annotations remain incomplete and methods to define them continue to evolve. In contrast to protein-coding genes, lncRNA gene annotations tend to have poorly defined boundaries, as judged by their lack of characteristic hallmarks of transcription initiation and termination[8]. While computational methods can provide some guidance[12], accurate gene annotation requires the use of high-confidence transcriptomic evidence, such as sequencing of full-length cDNA[13]. Until a few years ago, only low-depth techniques, such as Sanger sequencing of expressed sequence tags (ESTs)[14], were used. Recent advances in high-throughput cDNA sequencing technology, that is, RNA-seq[15,16], have provided deep sampling of the human transcriptome[17]. When used in the context of gene annotation, however, these techniques still exhibit limitations due to the necessary compromise between read length and sequencing depth. Long-read sequencing (for example, Roche 454, Pacific Biosciences) can in principle provide close to full-length transcript sequences, but at low depth. Short-read RNA-Seq experiments (for example, Illumina Hi-Seq) routinely produce hundreds of millions of reads. However, such reads are far shorter than a typical mRNA or lncRNA transcript, which severely hampers accurate full-length isoform assembly[18]. In summary, current non-targeted, conventional cDNA sequencing methods are ineffective for reading the full dynamic range of transcript expression in the cell. This means that low-expressed transcripts, that is, the majority of lncRNAs, suffer from incomplete annotations.

Technical methods are being developed to address the problem of low-abundance transcript annotation. Recently, a high-throughput sequencing method called CaptureSeq was used for lncRNA characterization, in conjunction with Illumina short-read sequencing. It achieves targeted transcript enrichment by the hybridization of cDNA (derived from cellular RNA) to bead-linked oligonucleotide probes that are tiled and complementary to exons[19,20]. RNA CaptureSeq proved to be effective for the discovery of novel lowly expressed transcripts and allows for their quantification and assembly. However, this procedure has not been designed to specifically address the proper definition of 5′ and 3′ transcript ends, and as a result other methods are required for the precise experimental annotation of gene boundaries.

To improve the annotation of the boundaries of low-expressed genes, we coupled the widely used RACE technique (rapid amplification of cDNA ends[21]) to high-throughput sequencing— 'RACE-Seq'. In RACE-Seq, we carry out RACE with primers designed in targeted loci with the aim of producing cDNA sequences that reach the transcript termini. RACE products are then subjected to high-throughput long-read sequencing (for example, Roche 454). We here apply RACE-Seq to a selection of 398 lncRNA loci from the reference GENCODE v7 catalogue[7], most of them low-expressed and lacking typical gene boundary hallmarks. We discover 2,556 novel, manually curated rare isoforms. Two thirds of those extend their parent locus beyond their previously annotated boundaries, often reaching marks of transcription initiation and termination, such as CAGE tags and poly-adenylation sites. We found that both the sensitivity and specificity of RACE-Seq are greatly enhanced by the use of a second, nested set of priming oligonucleotides. Overall, we show that RACE-Seq is a highly efficient method, well-suited for both novel isoform discovery and gene boundary characterization.

## Results

**RACE-Seq general strategy and proof-of-concept**. The outline of the RACE-Seq procedure is depicted in Fig. 1. For each locus in a given set of annotated genes, 5′ and/or 3′ RACE primers are designed *in silico* along the transcript sequences so that the resulting RACE product has a suitable size for the long-read sequencing platform in use (see Methods). To limit off-target RACE amplification, it is beneficial to ignore primers exhibiting substantial sequence identity with any transcribed region in the genome other than their intended target (>80% identity in our test case). To increase further RACE specificity, a second 'nested' primer, placed as close as possible, downstream of the first one, can be designed using the same selection criteria as before. RACE reactions are then carried out in RNA extracted from the cellular samples of interest. Finally, RACE products are subsequently sequenced using a high-throughput long-read sequencing platform, and resulting reads are aligned and assembled into spliced transcripts on the genome.

As a proof-of-concept, we targeted 398 distinct lncRNA loci from the GENCODE v7 annotation[7,8], and performed RACE-Seq on a set of cDNA libraries from 7 human tissues (brain, heart, kidney, liver, lung, spleen and testis) known to cover a large fraction of the annotated human transcriptome[22]. We subdivided our set of target lncRNAs in two subsets, depending on whether their annotated 5′ end was supported by CAGE tags (as identified by the FANTOM project[3] ($N = 180$), or not ($N = 218$) (see Methods)). The RACE cDNA mixtures were then sequenced using the Roche 454 FLX+ platform. Reads obtained have an average length of ~600 bp. Sequenced reads were mapped to the genome using a combination of BLAT[23] and GMAP[24], and the resulting alignments were manually curated and incorporated into the GENCODE human gene set.

We obtained a first batch of RACE-Seq (referred to as 'standard RACE' below) using standard, non-nested RACE primers in each of the 398 targets. We then performed nested RACE on aliquots of the standard RACE reactions so as to improve the assay's sensitivity and specificity. In total, adding an extra pilot set of standard RACE libraries, we sequenced ~22 million reads in 40 RACE libraries (Supplementary Figs 1 and 2). We obtained at least one alignable RACE product for 94% of the 398 targeted loci, and discovered at least 1 novel, manually curated isoform for 343 of them.

**Novel gene boundaries**. With RACE-Seq, out of the 398 targeted lncRNAs, we extended 176 and 193 loci further in 5′ and 3′, respectively, and 131 in both directions (Fig. 2a, left panel). In total, the boundaries of 238 loci (60%) were expanded in either direction. These genomic extensions were accounted for by 752 and 848 distinct 5′ and 3′ RACE products, respectively (Fig. 2a, right panel). Eighty two novel transcripts extended their parent locus in both 5′ and 3′.

RACE-Seq was particularly successful in extending CAGE-unsupported loci: the median/mean genomic length of 5′ extensions were $+21/-8,479$ and $-376/-14,440$ (negative

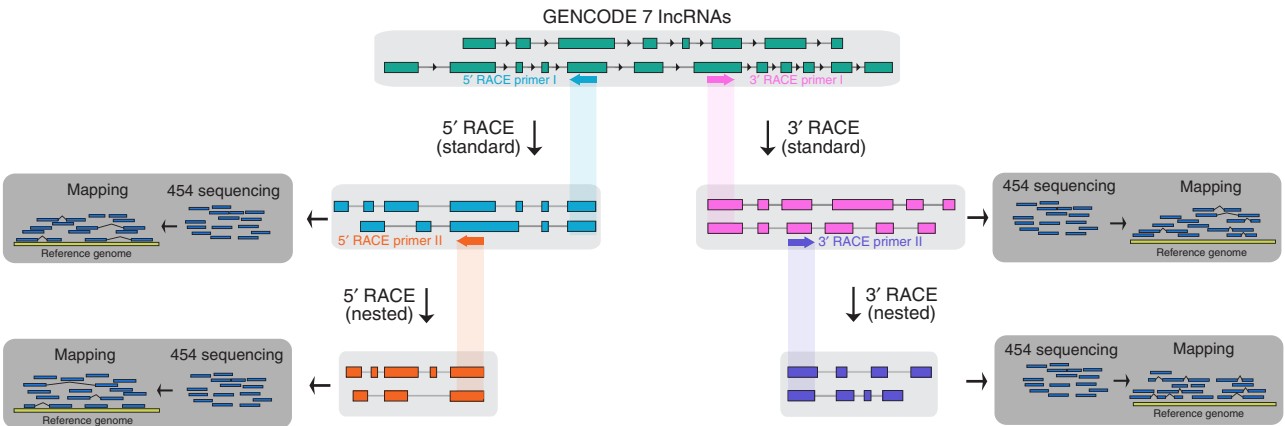

**Figure 1 | Schematic overview of RACE-seq.** Standard 5′ and 3′ RACE primers are designed to target exons of a gene and produce primary RACE products, which undergo a second round of RACE reactions using nested 5′ or 3′ RACE primers. Both standard and nested 5′ and 3′ RACE products are subjected to long-read sequencing, followed by mapping to the reference genome.

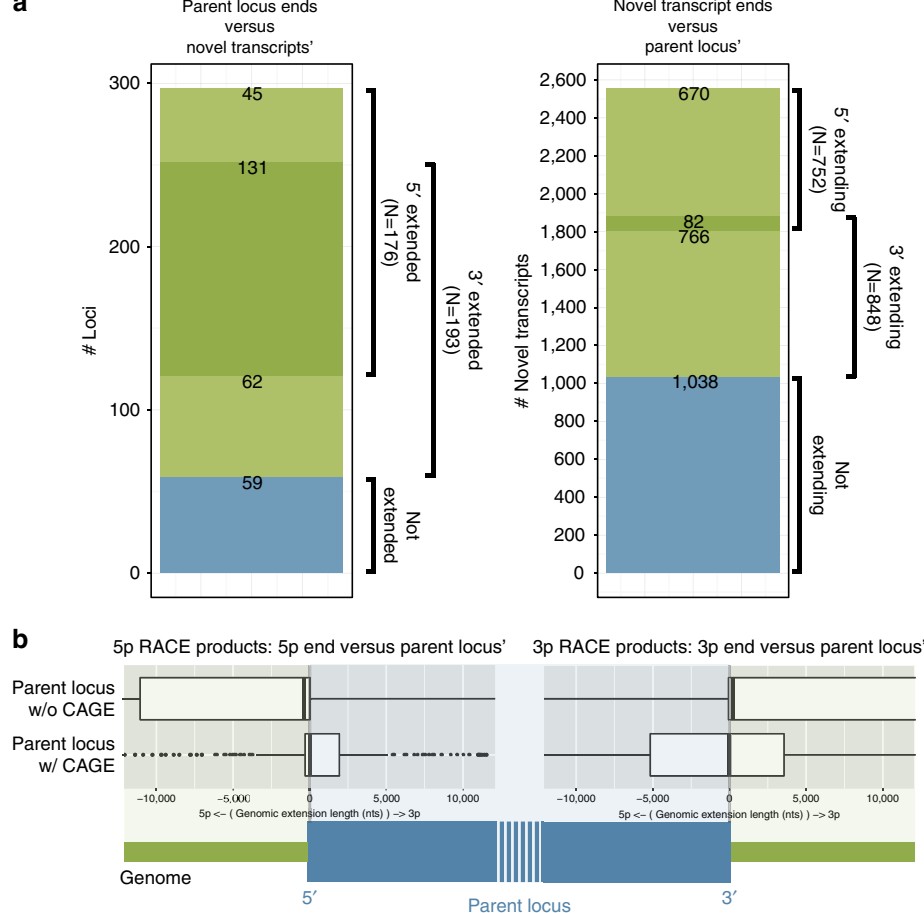

**Figure 2 | Locus extension and novel transcript boundaries.** (**a**) Venn diagrams depicting the proportion of loci (left panel) and transcripts (right panel) extended in 5′ and/or 3′ direction. (**b**) Novel boundaries for CAGE-supported (bottom box-plot) and CAGE-unsupported loci (top box-plot). A schematic depiction of a target locus is provided below the plots. The viewing range of the box plots is reduced ( − 10,000, 10,000 nucleotides) for clarity.

values represent novel transcription start sites (TSSs) upstream of the annotated locus), respectively, for CAGE-supported and unsupported loci (Fig. 2b, and example in Fig. 5b). Surprisingly, we observed a similar phenomenon at the 3′ end of targeted loci: the mean/median genomic length of 3′ extensions amounted to − 15/ − 526 and + 225/ + 8,518 (positive values correspond to novel transcription termination sites (TTSs) downstream of the

annotated locus′), respectively, for CAGE-supported and unsupported loci. We speculate that this observation is due to the pre-RACE-Seq GENCODE set being mostly based on oligo-dT-primed ESTs, which tend to cover preferentially the 3′ end of transcripts. As a consequence of this bias, a transcript model that is complete at its 5′ end (that is, CAGE-supported) is also likely to be complete at its 3′ end, which is consistent with our results.

**Table 1 | Comparison of various TTS data sets with Merck PolyA-Seq peaks.**

| Data set | Total #TTS | #TTS close to a polyA-Seq tag (±100 nts) | % TTS close to a polyA-Seq tag (±100 nts) |
|---|---|---|---|
| Targets (pre-RACE) | 535 | 83 | 16% |
| Targets updated (post-RACE) | 1,027 | 99 | 10% |
| Protein coding | 17,940 | 7,019 | 39% |
| lncRNAs | 12,556 | 2,223 | 18% |

lncRNA, long non-coding RNA; RACE, rapid amplification of cDNA ends; TTS, transcription termination site.
Statistics are also reported for the full sets of GENCODE-annotated protein-coding genes and lncRNAs for reference.

**Table 2 | Comparison of pre- and post-RACE TTSs data sets with polyA peaks called using our RACE-Seq data.**

| Data set | Total #TTS | #TTS close to a RACE-Seq inferred polyA-Seq tag (±100 nts) | %TTS close to a RACE-Seq inferred polyA-Seq tag (±100 nts) |
|---|---|---|---|
| Targets (pre-RACE) | 535 | 206 | 39% |
| Targets updated (post-RACE) | 1,027 | 321 | 31% |

RACE, rapid amplification of cDNA ends; TTS, transcription termination site.

In addition, we observed that when novel TSSs were discovered in CAGE-supported loci, they were much more likely to be supported by another CAGE peak than CAGE-unsupported loci (74% versus 56% CAGE support, see Supplementary Fig. 3). Overall, the experiment uncovered 873 non-redundant TSSs, of which 615 were previously unknown—including 252 (41%) that were CAGE-supported (Supplementary Table 1).

We also assessed the accuracy of the newly annotated TTSs by comparing them with experimentally established poly-adenylation (polyA) sites (Merck PolyA-Seq data sets[25]). The overall proportion of TTS within 100 nucleotides of a PolyA-Seq tag slightly decreased from 16% pre-RACE to 10% post-RACE (Table 1). Yet, the raw count of TTSs supported by PolyA-Seq was improved after RACE-Seq, albeit very marginally (from 83 to 99 PolyA-Seq-supported TTSs). In addition, we identified polyA sites ourselves by searching for non-templated polyA/T tails in partially mapped 3′ RACE-Seq reads. Using this method, we were able to precisely map 1,212 distinct polyA sites near our targets, and compared those with the 3′ ends of our transcript set. We observed a much higher number of TTSs in the near vicinity (±100 nucleotides) of these sites (206 and 321 pre-RACE and post-RACE TTSs, respectively) (Table 2). This indicates that the low Merck PolyA-Seq coverage of our TTSs is probably due to the limited depth and tissue coverage of PolyA-Seq compared with our RACE-Seq data.

**On-target enrichment and sensitivity of RACE-Seq.** Since (1) RACE operates with only one internal oligonucleotide primer, and (2) our targeted genes are very lowly expressed ones, we expected this experiment to yield a high number of off-target products. We found that, on average across all samples, 94% of uniquely mapped sequencing reads overlapped GENCODE v7 genic regions (Supplementary Fig. 4), indicating insignificant genomic contamination of our cDNA libraries. The vast majority of reads arose from annotated genic regions, and 3.9% of them, on average, fell within the targeted locus boundaries when using standard RACE (Fig. 3a and Supplementary Table 2). This corresponds to a 3.1-fold enrichment of reads originating from transcripts compared with untargeted sequencing (as estimated using GTEx RNA-Seq data in matched tissues, see Methods). In contrast, nested RACE yielded an average of 36.4% on-target reads across all tissues (that is, a 9.5-fold increase in specificity compared with standard RACE, and 29.2-fold over expected in

untargeted sequencing), allowing much deeper sequencing of the target loci. Similarly, the number of non-targeted loci producing reads decreased 2.2-fold when using nested RACE (on average, 5,025 amplified non-targeted loci in standard RACE, versus 2,332 in nested RACE, see Supplementary Fig. 5).

The total number of loci successfully amplified by RACE-Seq was 374 (94%, regardless of RACE direction), 326 (82%) and 341 (86%) for 3′ RACE and 5′ RACE, respectively (Supplementary Table 3). When further assessing the sensitivity of RACE-Seq, we noticed the benefits of nested RACE over standard RACE again. Overall, 12.5% (351 versus 312) more targets could be amplified in nested RACE-Seq than in standard RACE-Seq. The majority of positive loci (289, that is, 73% of the total) were detected in both nested and standard RACE, and only 23 of them were positive in standard RACE only (Supplementary Fig. 6).

In each individual tissue, nested RACE-Seq always out-performed standard RACE-Seq (Fig. 3b). The median number of positive targets was 49 (12%) across all standard RACE-Seq experiments, and 130 (33%) across the nested ones. The difference in sensitivity between nested and standard RACE samples was particularly remarkable in the kidney 3′ RACE samples (36% versus 8% success rate, respectively), and less noticeable in more transcriptionally complex tissues, such as testis (3′ RACE; 58 versus 50% success rate, respectively). We attribute the nested sets′ sensitivity improvements to its better specificity, which, by limiting the number of off-target reads, leads to a deeper sampling of the targeted transcripts. Taken globally, these results indicate that, as one could expect, nested RACE-Seq is far more informative that standard RACE-Seq, that is, surpasses it in both sensitivity and specificity terms.

**Novel isoforms in targeted regions and tissue origin**. We extended 176 lncRNA loci at the 5′ end and 193 loci at the 3′ end out of the total of 398 loci targeted for extension from GENCODE v7 (Fig. 2a). After extension, re-annotation and loci merging where necessary, the total number of lncRNA loci was reduced to 343. One putative lncRNA locus (OTTHUMG00000009351), when extended, was found to bear coding potential as it was extended to overlap the *LRRC7* (Leucine-rich repeat containing) coding locus, thus its biotype was changed to protein-coding. About 57 transcripts were merged into existing protein-coding loci (see example in Supplementary Fig. 7, where a putative lncRNA is re-annotated to be part of the

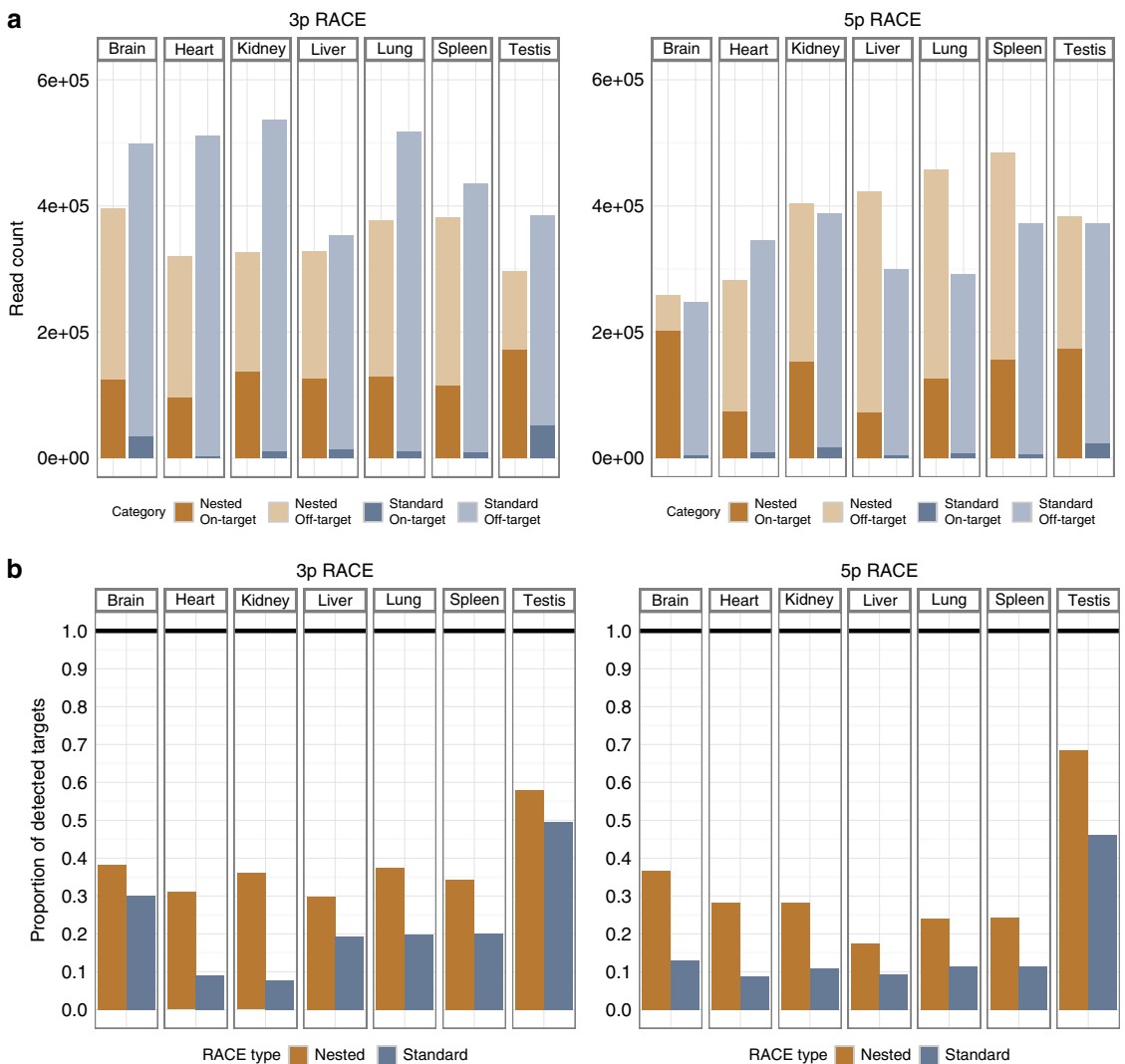

**Figure 3 | On-target RACE enrichment and RACE-Seq specificity.** (**a**) Number of RACE-Seq reads falling into exonic regions of targeted genes (dark shades) and outside of them (light shades), after using standard (blue) and nested (orange) 5′ and 3′ RACE. (**b**) Proportion of targets detected by standard (blue) and nested (orange) 5′ and 3′ RACE-Seq.

**Table 3 | Table summarizing basic annotation statistics before and after RACE-Seq.**

| Data set | #loci | #transcripts | #transcripts per locus | #exons (all) | #exons (unique) | #exons per transcript |
|---|---|---|---|---|---|---|
| Targets (pre-RACE) | 398 | 597 | 1.5 | 1,889 | 1,695 | 3.2 |
| Targets updated (post-RACE) | 343 | 2,556 | 7.5 | 10,139 | 5,326 (4,626) | 4.0 |

RACE, rapid amplification of cDNA ends.
Unique exons are those having distinct coordinates on the genome. The number of previously unannotated unique exons is indicated between parentheses in the penultimate column.

*PIGL* (phosphatidylinositol glycan anchor biosynthesis, class L) locus using RACE-Seq read data). The number of alternatively spliced variants generated by the 5′ and 3′ RACE increased by >4 fold from 597 to 2,556 (Table 3), and the median length of the transcripts slightly increased from 623 to 704, although not significantly ($P = 0.7$, Wilcoxon rank sum test with continuity correction) (Fig. 4a). It should be mentioned that RACE, by design, does not produce full-length, TSS-to-TTS transcripts. This is because RACE products, by definition, start at their originating primer's position along the targeted transcript. Therefore, we speculate that the length of post-RACE transcripts is heavily underestimated.

The average number of transcripts per locus increased from 1.5 (597/398) pre-RACE to 7.5 (2,556/343) post-RACE (Table 3 and Fig. 4b). The total number of splice junctions increased from 1,093 pre-RACE to 3,085 post-RACE (Table 4). One lncRNA, *PCB1-AS1* (OTTHUMG00000153728), antisense to *PCBP1*, had the highest number of alternatively spliced transcripts, increasing from 40 transcripts pre-RACE to 170. The function of this lncRNA is currently unknown, however, the *PCBP1* protein is known to act as a translational coactivator[26] and mediate the degradation of mitochondrial antiviral signals. Interestingly, *PCB1-AS1* was already highlighted by Derrien *et al.*[8] as the most alternatively spliced lncRNA gene in the GENCODE v7

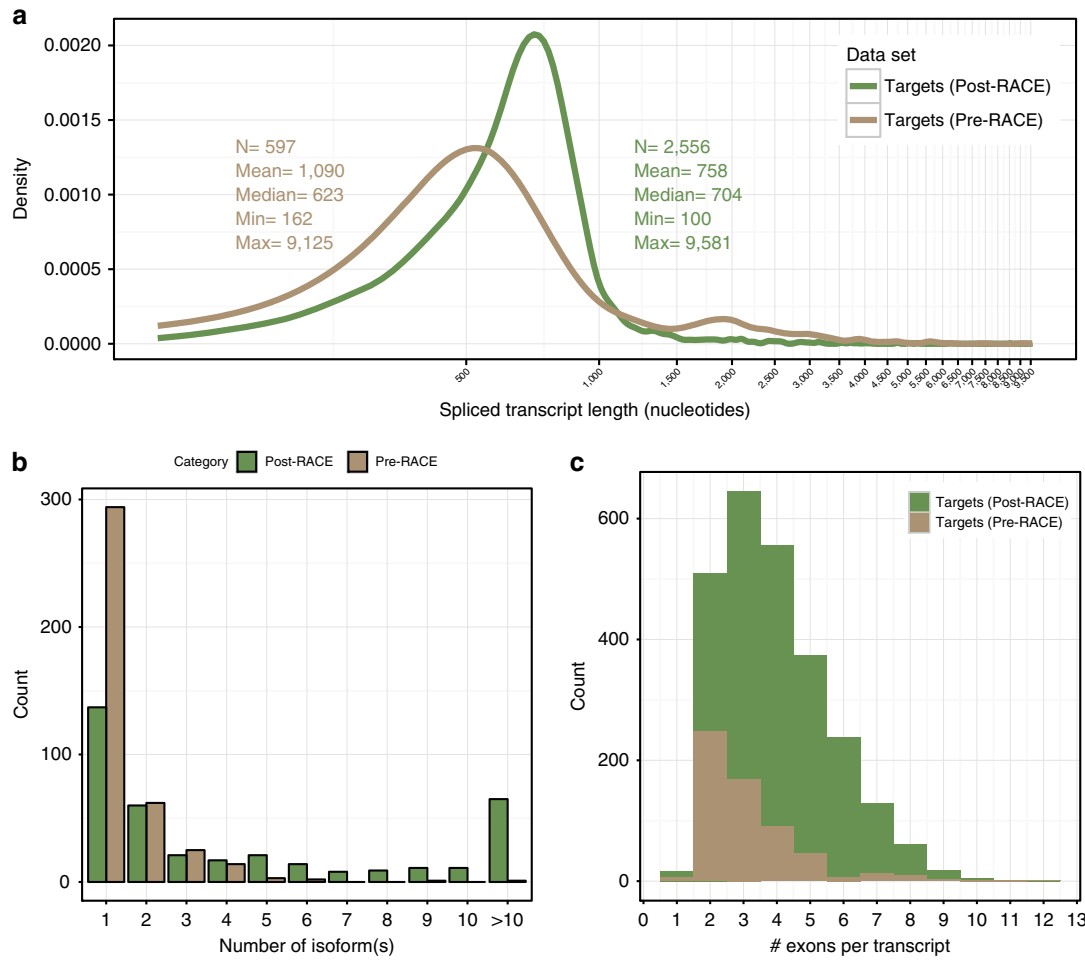

**Figure 4 | New isoform discovery and annotation.** (**a**) Length distribution of spliced transcripts (logarithmic scale) for pre- (brown) and post-RACE-Seq (green) targets. (**b**) Distribution of the number of alternatively spliced isoforms per pre- (brown) and post-RACE-Seq (green) targeted gene locus. (**c**) Exon count distribution in pre- (brown) and post-RACE-Seq (green) transcripts.

**Table 4 | Proportion of annotated splice junctions in pre- and post-RACE-Seq targets supported by short-read ENCODE or GTEx RNA-Seq data.**

| Data set | Total #unique splice junctions | #supported by ENCODE or GTEx RNA-Seq | %supported by ENCODE or GTEx RNA-Seq |
|---|---|---|---|
| Targets (pre-RACE) | 1,093 | 771 | 71% |
| Targets updated (post-RACE) | 3,085* | 975 | 31% |
| Protein coding | 82,627 | 74,090 | 90% |
| lncRNAs | 24,133 | 16,937 | 67% |

lncRNA, long non-coding RNA; RACE, rapid amplification of cDNA ends.
Both data sets are derived from conventional, unbiased sequencing experiments. ('*' represents novel introns only).

catalogue. Figure 5a shows a common occurrence in the annotation where two separate lncRNA loci have been extended to produce one larger new locus (*LINC01246*) with over 50 new spliced transcripts.

The majority, 63% ($N = 1,618$), of the 2,556 RACE-Seq derived transcripts were from testis and 20% ($N = 516$) from brain (Supplementary Fig. 8). The rest of the tissues (heart, kidney, liver, lung and spleen) contributed ~13% of novel transcripts. Many genes that appeared to be extensively alternatively spliced (>25 transcripts) such as *TEX1*, *LINC0069* and *LAMTOR5-AS*, are detected in all 7 tissues examined.

In total, 4,626 novel exons were discovered, bringing the average number of exons per transcript from 3.2 pre-RACE-Seq to 4.0 post-RACE-Seq. Derrien *et al.*[8] made the striking observation that lncRNAs have a very strong bias towards two-exon structures and exhibit less alternatively spliced isoforms per locus compared with protein-coding genes. Our results suggest that these are artifacts arising from inaccurate annotation of lncRNA transcript structures, since the biases towards both two-exon transcripts and isoform-poor genes disappear in the post-RACE-Seq transcripts (Fig. 4b,c).

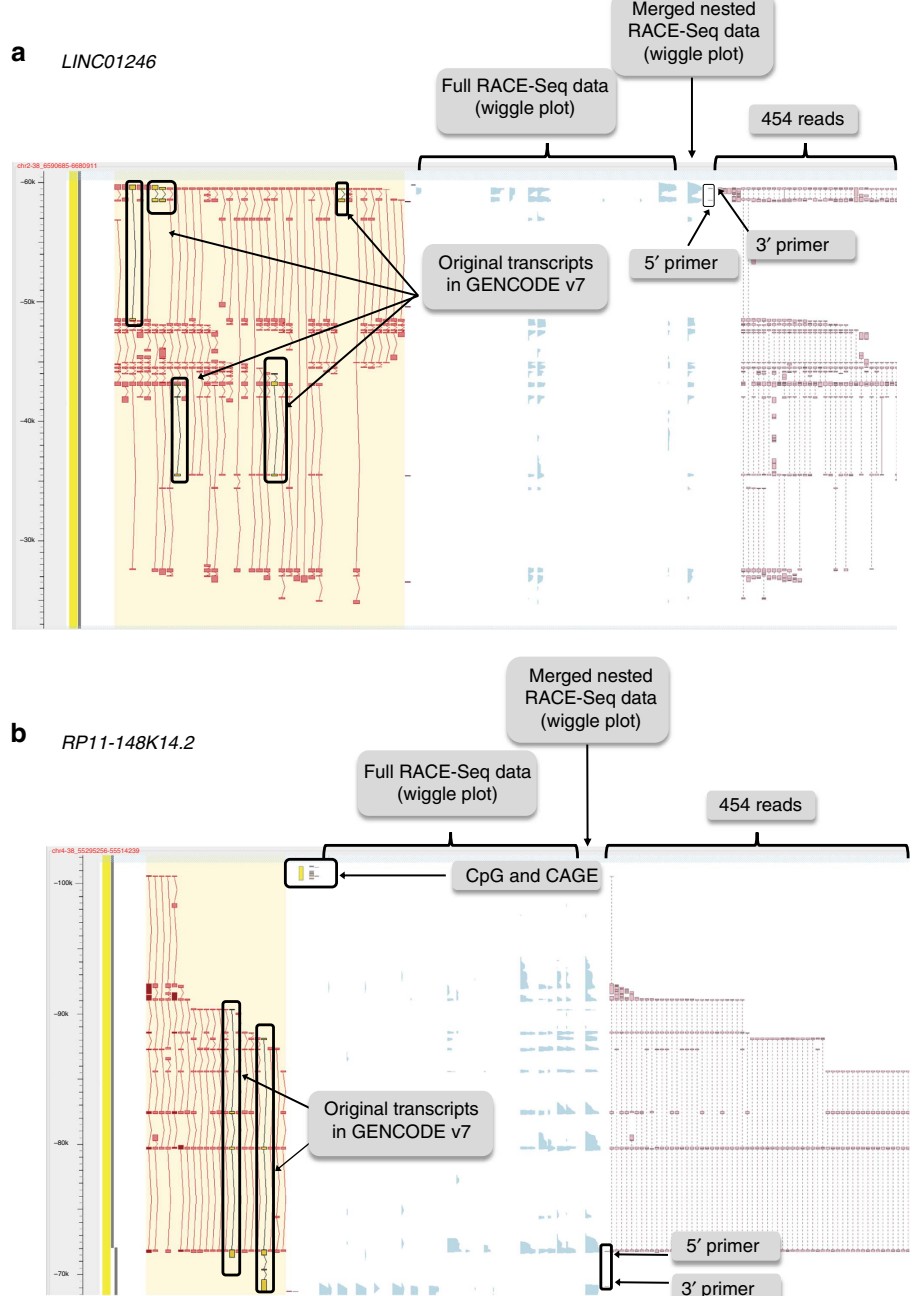

**Figure 5 | Locus examples.** (**a**) Two separate loci were merged into one larger locus (LINC01246). This example illustrates the large number of alternative splicing events found using the RACE-Seq approach. The red filled transcripts (far left) indicate the manual annotation models built from the 454 reads (far right in pink), as visualized in the ZMap browser (http://www.sanger.ac.uk/science/tools/zmap). (**b**) RACE-Seq reads (far right, in pink) establishes the Transcriptional start site (TSS) of an existing incomplete lincRNA, by extending the 5' end of the gene to a CpG island (yellow box) and is also supported by FANTOM5 CAGE data (small pink boxes).

**Comparison with other transcriptome sequencing methods**. To further evaluate RACE-Seq, we compared its performance with other transcriptome sequencing methods. First, we used non-targeted, conventional RNA sequencing data generated by the GTEx[11] and ENCODE[6] consortia. We analysed the GTEx pilot data freeze, which consists of RNA-seq data collected from 1,641 samples from 175 human individuals, representing up to 43 tissues per individual (29 solid organ tissues, 11 brain regions, whole blood and 2 cell lines). GTEx RNA-Seq samples were sequenced to an average 80 million of pair-end Illumina reads ($2 \times 76$ bp) per sample. The ENCODE data set is smaller (55 human cell lines and 104 samples), but on the other hand much more deeply sequenced (200–250 million pair-end reads ($2 \times 100$ or $2 \times 76$ per sample)). We found that 71% of pre-RACE-Seq splice junctions from the targeted loci were supported by short-read Illumina ENCODE or GTEx RNA-Seq data. This proportion dropped to 31% when looking only into novel splice junctions found in transcripts discovered through RACE-Seq (Table 4). This result strongly suggests that the coverage of weakly expressed novel transcripts in non-targeted, conventional RNA-Seq experiments is shallow. It is also reflected by the proportion of overall support of splice junctions for

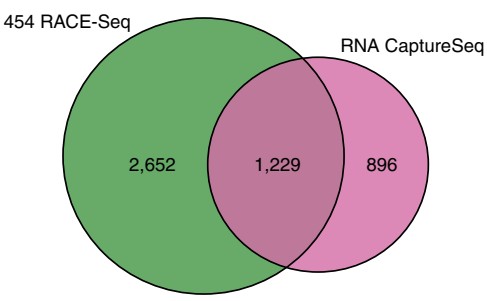

**Figure 6 | RACE-Seq performance compared with CaptureSeq.** Venn diagram indicating the number of annotated and unannotated on-target (± 5 kb) splice junctions discovered by RNA CaptureSeq and RACE-seq. Only the top 25% splice junctions with canonical splice sites ranked by read coverage were included in this analysis (see Methods).

protein-coding and lncRNA genes from GENCODE v7 by ENCODE or GTEx RNA-Seq data. Almost all splice junctions from annotated protein-coding loci show short-read support, while it is the case for <70% of lncRNAs.

To fully assess the performance of RACE-Seq, we compared our results with another targeted RNA sequencing method, capture sequencing (RNA CaptureSeq). RNA CaptureSeq enhances coverage of weakly expressed transcripts by focusing sequencing on genes of interest, thus enabling deeper sampling of low-abundance isoforms[19,20]. We analysed a subset of seven matching tissues from RNA CaptureSeq data set generated for lncRNAs profiling across 20 different human tissues by Clark *et al.*[20]. The support rate for both pre- and post-RACE transcripts is much higher compared with conventional RNA-Seq experiments, with 83% and 60% of splice junctions supported by RNA CaptureSeq data, respectively (Supplementary Table 4). To investigate whether RACE-Seq provides deeper interrogation of transcriptional events, we compared the set of splice junctions produced by each method within boundaries (± 5 kb) of 366 loci targeted by both studies. To compensate for differences in sequencing depth, we considered only the top quartile of canonical splice junctions, as ranked by read coverage, in each data set. RNA CaptureSeq enabled detection of 2,125 splice junctions, while RACE-Seq of 3,881 (83% more). Moreover, 1,229 splice junctions were supported by both methods, which constituted 60% of the total number of splice junctions seen in RNA CaptureSeq and roughly 30% from RACE-Seq (Fig. 6). Both techniques produce splice junctions uniformly distributed across targeted loci, including 5′ and 3′ ends (Supplementary Fig. 9). It is important to stress that the isoform discovery rate of both methods is expected to be negatively correlated with the number of targeted genes (16,453 in the study by Clark *et al.*[20], 398 in the present one), owing to the limited sequencing depth they rely on. These differences are not fully accounted for in our analysis, and may therefore favour our method over CaptureSeq in this comparison.

In addition, we used the read data by Clark *et al.*[20] from equivalent tissues to build CaptureSeq Cufflinks[27] transcript models overlapping our target genes, and mapped their corresponding 5′ ends. We derived a total of 343 non-redundant TSSs from this set, of which only 70 (20%, including 37 supported by FANTOM5 CAGE data) were previously unknown according to GENCODE (Supplementary Table 1). When compared with the output of RACE-Seq (873 TSSs, including 615 novel ones, see section above), this highlights the superiority of this latter technique at uncovering novel TSSs in comparison with CaptureSeq.

## Discussion

Increased resolution in available technologies to monitor cellular transcriptomes have recently unveiled a plethora of RNA species beyond mRNAs. Among them, some lncRNAs have been shown to play important roles in cell function[28,29]. LncRNAs have characteristic tissue specificity and low-expression levels, which makes them challenging to annotate. While mRNAs, as well as some small RNA families, exhibit sequence and/or structural constraints that can be employed by computational methods to facilitate their identification and annotation, such constraints are mostly absent among lncRNAs[30]. There is indeed strong evidence that the exonic structure and the transcript termini of lncRNAs are not as well-annotated as those of protein-coding genes. For instance, only 15% of them have ENCODE CAGE data support at their 5′ end compared with 55% of protein-coding loci, according to a 2012 study[8].

Here we introduced the RACE-Seq methodology and used it to enhance lncRNA annotation. The idea of combining RACE with high-throughput sequencing was previously described by Olivarius *et al.*[31]. However, this study presented only 5′ RACE analysis of 17 protein-coding genes and compared single short-read Illumina sequencing with Sanger sequencing, and thus did not fully explore the high-throughput potential of this approach. In contrast, we tested the approach on a set of almost 400 human lncRNAs in 7 tissues, with both 5′ and 3′ RACE, and uncovered many previously unannotated transcripts. We increased the number of transcripts per lncRNA locus from 1.5 to 7.5 (see Table 3), and extended the 5′ and/or 3′ boundaries of the loci in 60% of the cases. The CAGE coverage of TSSs within the targeted genes increased by 28% at the end of the experiment—from 180 to (180 + 50 =) 230 CAGE-supported loci (Supplementary Fig. 3). Particularly useful was the usage of nested RACE-Seq, which lead to a 2.2-fold reduction in the number of detected off-target loci (Supplementary Fig. 5) compared with standard RACE-Seq.

While RACE-Seq leads to the identification of many novel transcripts, still only about 50% of the transcripts are on average full-length in a given locus. This could be improved by replacing the 454 technology, which has an average read length of 600 bp, with a longer-read sequencing technology, such as PacBio or Nanopore[32]. The sensitivity of RACE-Seq coupled with longer reads will facilitate automatic assembly of individual transcripts, which has proved problematic and inaccurate when using shorter reads[18], and it will lead to improved annotations. Indeed, we used the very large collection of short-read RNA-Seq samples from multiple tissues compiled by the GTEx project[11], and found that only 31% of the targeted lncRNA splice junctions could be detected in this data set. This highlights that conventional, unbiased short-read RNA-Seq suffers from a limited sampling capacity given the large dynamic range of transcript abundances within the cell.

To alleviate, in part, the poor sensitivity of unbiased methods, strategies that target specific genomic regions have already been developed. Notably CaptureSeq[19,20] uses oligonucleotide capture to perform short RNA-Seq in RNA populations enriched for selected loci. Still, we found that 40% of the RACE-Seq splice junctions are not observed in the output of CaptureSeq, and that although only 7 tissues were used in RACE-Seq this resulted in the discovery of a larger number of transcripts (6.6 on average per locus) than CaptureSeq (3.6 transcripts per locus on average) even when 20 tissues were employed. This highlights the importance of using longer reads and shows that RACE-Seq is very efficient to target specific gene classes, such as lncRNAs to uncover deep transcriptional complexity. Moreover, RACE-Seq has the advantage over CaptureSeq that it solves much more

accurately, and with more sensitivity the 5′ and 3′ end of transcripts.

The vast transcriptional complexity uncovered through RACE-Seq could reflect functional non-coding RNAs[33], or alternatively may be a result of experimental artifacts, and transcriptional noise[34]. The fact that many of the extensively alternatively spliced (>25 transcripts) loci such as TEX1, LINC0069 and LAMTOR5-AS show expression in all 7 tissues examined indicates that this complexity is not due to experimental artifacts, and could instead be an indication of the vast functional potential of non-coding RNAs[33]. However, the amount of biological 'noise' that exists within the transcriptome remains a source of much debate[34,35], and other methods will be required to rigorously establish the functionality of these transcripts.

Recently, Tilgner et al.[36] used a new sequencing method, Synthetic long-read RNA sequencing (SLR-RNA-Seq) in which small pools of full-length cDNAs are fragmented and sequenced using small-read-sequencing, and then re-assembled. Since Tilgner et al.[36] also examined the transcriptome of human brain tissue, we examined the data to investigate if any of the targeted lncRNA loci were detected by SLR-RNA-Seq. Around 37% ($N = 149$) of our targeted lncRNA genes were covered by reads, however, only 9% or RACE-Seq splice junctions were detected by SLR-RNA-Seq reads (see Supplementary Fig. 10 and Supplementary Table 5). This alternative method of deep short-read sequencing, combined with targeted, nested RACE-Seq, could potentially provide a cheaper alternative to more expensive longer-read sequencing.

## Methods

**Target selection and primer design.** The experiment was designed using a fully automatized pipeline, which contents are available on request. Illumina HBM (Human Body Map 2.0) RNA-Seq data was used and RPKMs (reads per kilobase of exon per million mapped reads) for all GENCODE v7 lncRNAs were calculated. We selected lncRNAs that were expressed in at least one HBM experiment with an RPKM >5 and that were lacking CAGE/PET support in ENCODE cell line experiments[6,17]. The spliced RNA sequences for the top 398 lncRNAs, ranked by mean RPKM across cell lines, were extracted and used as input for primer design. At the time of the experimental design, CAGE data on matched tissues were not available, therefore we had to rely on ENCODE CAGE experiments, performed on various cell lines, all quite distinct from our set of tissues. On the public release of matched tissue CAGE data from the FANTOM5 consortium[3], we re-calculated CAGE support of the 398 RACE-Seq-targeted loci. We found that, in fact, 180 of them had at least 1 CAGE tag in their vicinity (± 50 nucleotides, on the same strand) in at least 1 of the matched FANTOM5 tissues.

Non-specific regions within the candidate sequences were masked to avoid off-target RACE products. These regions were established by aligning candidate sequences against all GENCODE v7 transcript sequences using the BLAST program[37]. Regions having >80% sequence similarity to any GENCODE v7 transcript from a distinct locus were hard-masked. Only stranded overlap was considered. We then generated for each candidate transcript, all possible 5′ and 3′ RACE primers using primer3 with the following parameters: PRIMER_INTERNAL_OPT_SIZE = 25, PRIMER_INTERNAL_MIN_SIZE = 23, PRIMER_INTERNAL_MAX_SIZE = 27, PRIMER_INTERNAL_OPT_TM = 70.0, PRIMER_INTERNAL_MIN_TM = 68.0, PRIMER_INTERNAL_MAX_TM = 72.0, PRIMER_INTERNAL_MIN_GC = 50, PRIMER_INTERNAL_MAX_GC = 70, PRIMER_INTERNAL_OPT_GC_PERCENT = 60.

In total, we could design 3′ and 5′ RACE for all 398 targets in standard RACE. 361 and 367 nested primers could be designed for 3′/5′ RACE, respectively.

The full list of RACE primer sequences, together with their corresponding transcript targets and mean RPKM, is provided as a tab-separated file in the Supplementary Data section.

**RACE reactions.** Nested and non-nested 5′ and 3′ RACE products were obtained using the Clontech SMART RACE cDNA Amplification kit and the Advantage 2 Proofreading Polymerase PCR kit (Clontech Laboratories, Mountain View, CA, USA, catalogue no. 634914) according to the manufacturer's instructions. PolyA + RNA from a panel of seven human tissues was used (brain, heart, kidney, liver, lung, spleen and testis), all from Clontech Laboratories. RACE- and nested RACE-specific primers were synthesized by Life Technologies Europe BV and were diluted to a final concentration of 200 nM. Each RACE reaction was performed in an independent well on a 384 well plate, and PCRs were done using liquid-handling robots.

Double-stranded cDNA synthesis, adaptor ligations to the synthesized cDNA and RACE reactions were performed according to the manufacturers' instructions. Nested RACEs were performed with 0.5 μl of the initial RACEs in a final volume of 12.5 μl. The cycling parameters were: RACE 5 × (94 °C 30″, 70 °C 30″, 72 °C 3′), 5 × (94 °C 30″, 68 °C 30″, 72 °C 3′), 20 × (94 °C 30″, 66 °C 30″, 72 °C 3′); nested RACE 25 × (94 °C 30″, 68 °C 30″, 72 °C 3′). We then pooled by tissue, 2 μl of all nested RACE reactions, and pools were purified using Qiaquick PCR purification kit (Qiagen, CA, USA) before proceeding with 454 + library preparation.

**GS-FLX 454 + library preparation and sequencing.** cDNA RACE samples were analysed on a DNA 7,500 Chip (2,100 Bioanalyzer, Agilent Technologies Inc, Santa Clara, CA, USA) to assess fragment size and sample integrity. For samples with a mean fragment size of ≥2 Kbp, 1 μg of material is subjected to nebulization and then used to prepare a rapid ligation (RL) genomic shotgun library using the Rapid Library Preparation Method Manual (GS FLX + Series-XL +, May 2011, Roche 454 Life Sciences). For RACE samples with a mean fragment size smaller than 2 Kbp the nebulization step is avoided, starting library preparation directly with 800 ng of material. A modification was introduced in the small fragment removal of this library preparation, to allow only for the removal of fragments under 400 bp instead of fragments under 650 bp. Then, the quality of the RL libraries was assessed by running an aliquot of the library in a High Sensitivity Chip (2,100 Bioanalyzer). Library quantification was performed generating a RL standard curve and using a 96-well Plate fluorometer, according with the manufacturer's instructions, and using the Rapid Library Quantitation Calculator (www.454.com/my454).

Samples were titrated using the emPCR amplification Method Manual Lib-L SV (GS FLX + Series-XL +, May 2011) to know the optimal point of copies per bead (cpb) needed to obtain a 10% enriched beads. Then, a large volume emulsion PCR was performed using the emPCR amplification Method Manual Lib-L LV (GS FLX + Series-XL +, May 2011).

Sequencing was performed at the Genomic and Bioinformatics Platform of Andalusia (GBPA) using half 454-pyrosequencing plate per sample using a Roche 454 GS FLX + instrument, and GS FLX + reagents (Roche 454 Life Sciences). After the sequencing was finished, sequencing images were analysed using the Shotgun-pipeline to generate SFF files.

**Read pre-processing and mapping.** FASTQ files were extracted from SFF files by the program sffextract (http://bioinf.comav.upv.es/sff_extract/index.html). Cutadapt was used to remove adaptors, and reads shorter than 100 nts were filtered out (5′ RACE Adapter: 5′-CTAATACGACTCACTATAGGGCAAGCAGTGGTATCA ACGCAGAGTACT-3′, 3′ RACE Adapter: 5′-CTAATACGACTCACTATAGGGC AAGCAGTGGTATCAACGCAGATGACGCGGG)-3′.

Low quality nucleotides in 3′ end were hard-trimmed by calculating the mean quality of the last three nucleotides and removing bases progressively until reaching a mean quality >20 (Sanger scale).

Two different approaches where used to map the reads to the reference genome:

a) The Inchworm wrapper program[38] was used to run BLAT[23] (v35) and generate SAM files, by setting minimum per cent identity at 95% and considering just the best single hit per read reported by BLAT. Intron prediction by Inchworm is based on the presence of splice consensus sites in the ends of the gaps (gap size>20).

b) Using the GMAP program[24] (version 31 March 2013 with all parameters set to default except `--min-identity=0.95 --force-xs-dir -B 5 -t 5 -f samse $file --min-intronlength=30 --split-output`). Only unique mappings were considered in subsequent analyses.

Both BLAT and GMAP mappings were performed against the hg19 (GRCh37) assembly of the human genome.

**Coverage and on-target enrichment calculation.** Mapped reads were compared with annotated regions using the BEDtools suite[39] v2.17.0. Reads were considered on-target when they overlapped exonic regions of the targeted transcripts. We estimated the expected read coverage of transcripts in a typical non-targeted RNA-Seq experiment using GTEx[11] data in matched tissues (SRA accessions: SRR1403958, SRR1340617, SRR1314940, SRR1080294, SRR809807, SRR1069539 and SRR1458955). We then calculated the on-target enrichment achieved by RACE-Seq using the following formula:

$$\text{Enrichment} = R/E$$

Where

R = proportion (across tissues) of mapped RACE-Seq reads on-target
E = proportion (across tissues) of mapped GTEx RNA-Seq reads on-target. The results of the comparison between GTEx read coverage and both standard and nested RACE-Seq on the 398 targets are summarized in Supplementary Table 2.

**Transcripts manual annotation.** Manual annotation was performed according to GENCODE standards[7]. Briefly, imported BAM files (merged outputs of GMAP and BLAT) representing the aligned RNA-seq reads were displayed in our in-house annotation tool, ZMAP (http://www.sanger.ac.uk/science/tools/zmap). Splice sites and alignments for all non-redundant novel intron combinations and exon extensions were evaluated manually and, when confirmed, used to create new or

extend existing transcript models. While the target loci were lncRNAs, where warranted by the RNA-seq data, biotypes were modified (for example, from non-coding to coding if RNA-seq read joins lncRNA target to a coding gene).

**Characterization of novel transcript boundaries.** In this part of the analysis, we split all post-RACE-Seq transcript models into 5′ and 3′ RACE products. We reasoned that 5′ RACE product models may be anchored at their originating primer location at the 3′ end, hence obfuscating the global analysis of genuine transcript 3′ ends (and likewise for 3′ RACE products versus 5′ ends). We did so by assigning the most probable originating RACE direction (5′ or 3′) to each post-RACE GENCODE transcript model: a transcript extending an annotated locus further in 5′, and/or whose 3′ end started within 50 bps of a 5′ RACE primer (and on the opposite strand) was labelled a 5′ RACE product. Similarly, one extending an annotated locus further in 3′, and/or whose 5′ end started within 50 bps of a 3′ RACE primer (and on the same strand) was labelled a 3′ RACE product.

This resulted in 1,427 and 1,420 transcript objects likely produced by 5′ RACE and 3′ RACE that were used for TSS and TTS analysis, respectively. About 291 transcripts were shared between those 2 sets (that is, likely complete structures from 5′ to 3′). Overall, we could assign a probable RACE direction to 2,556 transcripts out of 2,641, that is, 97%.

**Transcription start sites.** All annotated 5′ RACE-Seq TSSs were clustered (that is, all TSSs on the same strand and <51 bps away were merged into one). About 615 out of these 873 clustered TSSs were considered novel, that is, they lied farther than 100 bps from any of the targeted GENCODE 7 transcript's TSSs. We compared TSSs against merged, tissue-matched CAGE data from the FANTOM5 consortium, and considered them CAGE-supported if a CAGE tag could be found on the same strand, within 50 bps on either side. Of the 615 novel TSSs, 252 (41%) were found to be CAGE supported.

TSSs uncovered with RACE-Seq were compared with the TSSs of their originating locus. Any RACE-Seq TSS upstream of the 5′-most TSS of its original, GENCODE 7-annotated locus, was labelled as 'extending', and the corresponding locus as '5′-extended'.

**Transcription termination sites.** All annotated 3′ RACE-Seq TTSs were clustered (that is, all TTSs on the same strand and <151 bps away were merged into one). The clustering distance was chosen longer than for TSSs because of the 'leakier' nature of transcription termination compared with transcription initiation. We compared TTSs against merged data from available matched tissues (brain, kidney, liver, muscle, testis) from the Merck PolyA-Seq set[25], as downloaded from the UCSC genome browser. We considered a TTS polyA-Seq-supported if a PolyA-Seq tag could be found on the same strand, within 100 bps on either side.

We also inferred polyA sites from RACE-Seq data. To do so, we selected mapped 3′ RACE reads and searched for characteristic non-templated stretches of >20 Ts or As (allowing for 10% mismatches) at their ends. About 1,212 non-redundant polyA sites were mapped, and compared with RACE-Seq TTSs in the same manner as for Merck PolyA-Seq sites.

TTSs uncovered with RACE-Seq were compared with the TTSs of their originating locus. Any RACE-Seq TTS downstream of the 3′-most TTS of its original, GENCODE 7-annotated locus, was labelled as 'extending', and the corresponding locus as '3′-extended'.

**Conventional unbiased short-read RNA-seq.** Integrative Pipeline for Splicing Analyses (IPSA, unpublished, https://github.com/pervouchine/ipsa) was employed to locate splice junctions from 1,641 GTEx, 104 ENCODE and 38 454-RACE-Seq bam files, respectively. Alignments for GTEx and ENCODE data sets were produced by each consortium using their respective official processing pipelines. IPSA was run with the default parameters except −entropy 3. The analysis of splice junction support was done by comparing the two lists of splice junctions: one consisting of GTEx or ENCODE splice junctions produced by IPSA, and a second containing splice junctions derived from GTF files of pre- or post-RACE transcripts.

**RNA CaptureSeq.** RNA CaptureSeq FASTQ files for seven tissues matched with RACE-seq experiment were downloaded from BioProject (PRJNA261251). The sequences were aligned to the reference human genome (GRCh37/hg19) using STAR[40] v 2.4.0, according to the instructions specified by Clark et al.[20]. Both standard and nested 454- RACE-seq sequences were also re-mapped using STAR v 2.4.0. The following non-default STAR parameters were applied: `--outSAMunmapped Within --alignSJDBoverhangMin 1 --outFilterType BySJout`. Again, IPSA (with the same parameters as those mentioned above) was run to produce the lists of annotated and unannotated splice junctions for both data sets. Support rate of splice junctions annotated by GENCODE for pre- and post-RACE loci (only those targeted by both studies) by RNA CaptureSeq and 454-RACE-Seq was investigated by intersecting those with the set of splice junctions detected by IPSA for each data set. Next, annotation-free splice junction analysis was performed to further compare RNA CaptureSeq and 454-RACE-Seq. This analysis was done by selecting annotated and unannotated

splice junctions with canonical splice sites, located within targeted loci boundaries (±5 kb) from the IPSA output. Next, splice junctions were ranked by read coverage and only those supported by top quartile read count were further analysed.

We assessed the TSSs discovered by CaptureSeq by re-building Cufflinks[27] transcript models from Clark et al.'s[20] provided BAM files on brain, heart, kidney, liver, lung, spleen and testis (GEO accession GSE61474). Reads falling on and in the vicinity (±500 bps) of our GENCODE 7 targeted genes were selected and fed to Cufflinks v.2.2.1 with all options set to default except '`--library-type fr-firststrand`', '`-u`' and using GENCODE 7 as a guide ('`-g`'). We derived TSSs from Cufflinks' output after selecting those 1,156 transcript models that overlapped RACE-Seq-targeted exons, had the 'full_read_support' GFF attribute set to 'yes' and an FPKM value >0 in any of the 7 tissues. The resulting 343 non-redundant TSSs were then processed the same way as RACE-Seqs (see section above and results in Supplementary Table 1).

**Synthetic long-read sequencing.** The sequences for 11 human brain samples produced using SLR-seq were downloaded from Sequence Read Archive (SRP049776) and aligned to the reference genome (GRCh37) using GMAP and parameters specified by Tilgner et al.[36]. An in-house developed pipeline was applied to produce the list of splice junctions from genomic alignments.

**Data availability.** All computer code is available from the authors upon request. Sequence data have been deposited in the European Nucleotide Archive (ENA) under accession number ERP012249. All curated novel isoforms were incorporated into the human GENCODE set (version 22 onwards). In addition, a data portal, including a UCSC track hub, is available at http://public-docs.crg.es/rguigo/Papers/2016_lagarde-uszczynska_RACE-Seq/.

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

## Acknowledgements

This work and publication were supported by the National Human Genome Research Institute of the National Institutes of Health (grant numbers U41HG007234, U41HG007000 and U54HG007004) and the Wellcome Trust (grant number WT098051). Work in laboratory of R.G. was supported by Awards Number U54HG0070, R01MH101814 and U41HG007234 from the National Human Genome Research Institute. The content is solely the responsibility of the authors and does not necessarily represent the official views of the National Institutes of Health. We acknowledge support of the Spanish Ministry of Economy and Competitiveness, 'Centro de Excelencia Severo Ochoa 2013–2017', SEV-2012-0208 and grant BIO2011-26205. We thank members of the Guigó laboratory for their valuable input when analysing data and writing the manuscript, in particular Rory Johnson, Dmitri Pervouchine and Sarah Djebali; Romina Garrido (CRG) for administrative assistance, and Roche for providing library preparation and sequencing reagents for the nested RACE experiments.

## Author contributions

R.G., J.H., J.D., J.S.-L., A.T., A.Re. and J.L. designed the experiment. A.Ru., F.J.L-D, A.T., J.D., J.S.-L., B.U.-R., J.M.G., E.T., J.M.M., C.S., L.W. and J.L. analysed the data. C.H. and J.C. performed the RACE amplification. A.V.-B. performed the 454 sequencing of the RACE products. J.L., B.U.-R., J.H. and R.G. wrote the manuscript.

## Additional information

**Competing financial interests:** The authors declare no competing financial interests.

