## [Peer Review File · Nature Communications]

Reviewers' comments:

Reviewer #1 (Remarks to the Author):

In the manuscript by Lagarde et al., the authors describe a method to annotate the structure of lncRNA. In this method, RACE is performed on a lncRNA using gene specific primers and the products are subjected to long read RNA sequencing. As a proof of concept the authors perform RACE-Seq on 398 lowly expressed transcripts and were able to identify several novel isoforms as well as better structure 5' and 3' ends. Although this study is interesting and will be useful to identify correct gene structure, it is still unclear how this technique will be utilized. It is not a high throughput technique as it requires making primers for each lncRNA manually. If a particular lncRNA (discovered from RNA-seq) seems biologically interesting, it would be far easier to perform RACE followed by Sanger sequencing.

Major Issues:

1. The authors choose 398 lncRNAs as an example to show the utility of RACE-Seq. The authors should provide a supplementary table with information about the 398 lncRNAs (Gene ID, mean RPKM etc) as well as primers used for RACE-seq.
2. The authors have not given the criteria on which they based the selection of the 398 lncRNAs.
3. The authors show that overall nested RACE-seq was more sensitive and specific than standard RACE-seq. It has already been very well established that nested RACE is always more sensitive and specific than standard RACE due to the nature of primer designing and second PCR reaction. It is unclear why the authors choose to perform such a comparison.
4. It would be interesting and more useful to the community if the authors choose a few published lncRNAs as examples and show in a figure how RACE-seq was able to better define the structure as well as 3' and 5' ends of these genes compared to what is already known about the transcript in ENCODE or any other resource.
5. Comparison of results of RACE-seq data generated by using pooled RNA with GTEx normal where individual samples are sequenced does not make any sense as the variation in transcript structure can be sample specific.
6. The authors have shown that some of the transcripts merged to form a single larger transcript. For example: locus OTTHUMG0000009351 extended and now shows coding potential. The authors did not do any validation of such claims.
7. Some nested primer results (6%) did not match with standard RACE primer results. Did the authors confirm the findings using other nested primers?
8. None of the figures are labelled properly. Even after asking the authors to label the figures, only the main figures were labelled. Supplementary figures were still left unlabeled. With no proper

labelling it is very difficult to know which figure the authors are referring to in the text. Also, there are no legends for the supplementary figures.

9. The formatting of the figures is also not done properly. Every figure has a different size making it very difficult to review. The authors are requested to properly format their manuscript before submission.

Reviewer #2 (Remarks to the Author):

Despite advances in transcriptomics, generating complete and accurate annotations of long noncoding RNAs (lncRNAs) is still difficult. Lagarde et al introduce RACE-seq, a novel method for targeted sequencing and annotation of otherwise difficult to study genes. RACE-seq is an extremely sensitive method for detecting and annotating transcripts and significantly outperforms other methods in the detection of lncRNA transcriptional start and end sites. Sensitive detection of transcript boundaries is the major advantage of RACE-seq, as other current methods cannot match it for the sensitive definition of transcript ends.

I found the study interesting and thought-provoking and the data and analysis generally sound. My only question is whether the study is "important" enough for Nature Comm.

Most of the text is clear and well written, however the first line of the abstract and the first half of the introduction need editing to improve the English.

Major points to address

- The manuscript could use more detail about the RACE-seq method.

An important aspect of the method that is not clear from the paper is whether RACE reactions are performed individually or as multiplexes. Was each RACE primer used in a separate reaction (i.e.: did targeting of ~400 loci require 400 RACE reactions) or were the RACE reactions performed with a pool of primers?

For others in the field considering adopting this technique an important aspect will be how easy/laborious is it and how much extra work is required as one scales up the number of gene loci targeted.

- The authors may wish emphasise their results about the greatly increased number of lncRNA isoforms per locus. Previous findings, (i.e.: Derrien et al) that there are few lncRNA isoforms per loci and much less than from expressed from coding loci, may be an artifact.

- The comparison between RACE-seq and CaptureSeq clearly shows RACEseq is superior at detecting transcript ends, especially when they are previously un-annotated. However I am not convinced by the analysis suggesting RACE-seq is more sensitive at transcript detection in general, as I don't think this is an apples-to-apples comparison. The RACE-seq is targeting ~400 loci, the compared

CaptureSeq design targeted every known lncRNA, >16000 loci and the sequencing was not saturating. A CaptureSeq design targeting only ~400 loci may well have shown similar sensitivity to RACE-seq.

I realise the authors have attempted to correct for this, but I don't see how this has created comparable data.

Minor comments

- Regarding the description of CaptureSeq in the introduction. The probes hybridise to cDNA not RNA.

Results

- "are designed in-silico, and picked along the transcript sequences". In this sentence, do you mean "positioned" not "picked"?

- the results state that primers with >95% identity to other transcribed region were not used. However the supplementary methods give a value of >80%. Can the authors please check this and correct if need-be.

- The authors state "Seventy- five novel transcripts extended their parent locus in both 5' and 3' over their entire length". I don't think stating "over their entire length" makes sense and I'm not sure the authors need to say this.

- The results in Figure 4b suggest that annotation via RACE-seq leads to lncRNAs being slightly longer than before. It would be useful if the authors could confirm this increase statistically.

- Figure 3a - please change the colours in the key to better represent those used in the figure.

- The supplementary word doc, still contains tracked comments from the first author, the authors may wish to remove this.

Reviewer #3 (Remarks to the Author):

A. Summary of the key results

The authors describe RACE-seq, which combines RACE with RNA sequencing to discover rare transcript isoforms and to accurately define the 5' and 3' ends of transcripts. Using this technique, they find that most targeted lncRNA loci are extended in the 5' and 3' direction, and discover more than 2,500 novel alternative transcripts.

B. Originality and interest: if not novel, please give references

The purpose of this work is highly interesting: lncRNA transcripts are poorly annotated, and in particular it is important to know the exact 5' end of the lncRNA, as it enables an analysis of regulatory control sequences in the proximal promoter region of the lncRNA.

However, the originality of the work is limited, as the following paper from 2009 already described the combination of 5' RACE with high-throughput sequencing:

Olivarius S, Plessy C, Carninci P: "High-throughput verification of transcriptional starting sites by Deep-RACE." *Biotechniques* 46(2): 130-132 (2009).

Advantages and disadvantages of RACE-Seq compared to Deep-RACE should be discussed in the manuscript.

C. Data & methodology: validity of approach, quality of data, quality of presentation

The quality of presentation could be improved. The English is sometimes rather awkward, and some of the figures are difficult to understand. In particular, it would be better to show Figure 2(a) as a Venn diagram, and to change the color legend of Figure 3(a). Also Figure 5(a) and (b) are hard to understand. Tables S4 and S5 should be shown in the main text, as the paragraph on the comparison to PolyA-Seq is quite difficult to understand in its current form.

I was confused by the sentence "This is probably due to the addition to many new short alternative variants, many of them anchored at their originating RACE primer location." What does this refer to?

The sentence "The function of this lncRNA ... and mediate the degradation of mitochondrial antiviral signals." does not add much to the theme of this manuscript, and can be dropped.

D. Appropriate use of statistics and treatment of uncertainties

No comments.

E. Conclusions: robustness, validity, reliability

I am surprised by the large fraction of off-target RACE amplification (though mitigated by the nested primer design). Is there any explanation for this?

Also I am concerned about the scalability of this approach. In spite of the high-throughput sequencing employed, only 398 lncRNA loci could be investigated.

F. Suggested improvements: experiments, data for possible revision

I can understand that RACE-Seq is successful at extending lncRNA loci at their 5' end if they are not supported by CAGE, but why is the same true for the 3' end?

Regarding the comparison to PolyA-Seq: I would expect that many lncRNA transcripts do not have a poly(A)-tail, which may explain the low coverage of the TTSs by PolyA-Seq.

G. References: appropriate credit to previous work?

The work by Olivarius et al. (see above) should be cited.

H. Clarity and context: lucidity of abstract/summary, appropriateness of abstract, introduction and conclusions

Other than the points raised above, the logical flow of the paper is good.

Reviewers' comments:
(Authors' responses are in green)

Reviewer #1 (Remarks to the Author):

In the manuscript by Lagarde et al., the authors describe a method to annotate the structure of lncRNA. In this method, RACE is performed on a lncRNA using gene specific primers and the products are subjected to long read RNA sequencing. As a proof of concept the authors perform RACE-Seq on 398 lowly expressed transcripts and were able to identify several novel isoforms as well as better structure 5' and 3' ends. Although this study is interesting and will be useful to identify correct gene structure, it is still unclear how this technique will be utilized. It is not a high throughput technique as it requires making primers for each lncRNA manually. If a particular lncRNA (discovered from RNA-seq) seems biologically interesting, it would be far easier to perform RACE followed by Sanger sequencing.

The assumptions made in this paragraph are incorrect. The supplementary text "Target Selection and Primer design" describes the fully automatic pipeline used to generate the primers. We only targeted 398 lncRNA not because the methodology is limited to this number, but rather because it was intended to be a pilot study. This methodology can be applied to hundreds of genes simultaneously, as (1) nested RACE-Seq is both highly sensitive and specific, as reported in our manuscript, and (2) PCR reactions can be done using liquid handling robots. It is also incorrect to suggest it is easier to perform RACE-Seq using Sanger sequencing than Next Generation sequencing, as Sanger sequencing would involve cloning the RACE products into a vector, transforming bacteria, selecting several hundreds—of clones and subsequently sequencing them instead of carrying out a single sequencing event of all RACE products.

Major Issues:

1. The authors choose 398 lncRNAs as an example to show the utility of RACE-Seq. The authors should provide a supplementary table with information about the 398 lncRNAs (Gene ID, mean RPKM etc) as well as primers used for RACE-seq.

This table has been added to the submission as Supplementary Table 1, and referred to in the Supplementary Text.

2. The authors have not given the criteria on which they based the selection of the 398 lncRNAs.

We thank the reviewer for this observation. While the criteria were presented in the first section of the supplementary text, entitled "Target Selection and Primer Design", we now explicitly summarize them in the introduction:

“We applied RACE-Seq on a selection of 398 low-expression lncRNA loci from the reference GENCODE v7 catalog⁷ that lacked typical landmarks of Cap Analysis of Gene Expression (CAGE) and Gene Identification Signature-Paired End diTagging²⁴ (GIS-PET) tags supporting their 5' and 3' end, respectively. “

In short, we targeted specifically lncRNAs with strong evidence that they were incompletely annotated at the 3' and 5' ends.

3. The authors show that overall nested RACE-seq was more sensitive and specific than standard RACE-seq. It has already been very well established that nested RACE is always more sensitive and specific than standard RACE due to the nature of primer designing and second PCR reaction. It is unclear why the authors choose to perform such a comparison.

We are puzzled by this comment by the reviewer. Obviously, we used nested RACE because it has been well established that it is always more sensitive and specific than standard RACE. However, nested RACE is also more costly and experimentally involved. Therefore, we wanted to evaluate whether in a high-throughput setting such as ours in which hundreds of RACE reactions need to be carried out in parallel, the extra effort involved in the second RACE reaction pays off in terms of transcript discovery. We believe that this is the right comparison to perform, and are surprised that the reviewer seems to disagree. In addition, we wanted to assess if we could specifically amplify lowly expressed lncRNAs without a high degree of noise from protein coding genes, while doing nested RACE. However as lncRNAs are very low expressed we wanted to assess if single RACE experiments provided enough amplification of the transcripts to avoid a second amplification.

4. It would be interesting and more useful to the community if the authors choose a few published lncRNAs as examples and show in a figure how RACE-seq was able to better define the structure as well as 3' and 5' ends of these genes compared to what is already known about the transcript in ENCODE or any other resource.

By design, our study focuses on poorly annotated lncRNA genes, and as such, it is devoid of well-studied, published loci. Figures 5a and 5b show the structure of the transcripts prior to the RACE-seq step (labelled original transcripts). We have relabelled the diagram “original transcripts in GENCODE v7 “ to make it clear this is the annotation used by Encode and ENSEMBL resources, and the extended transcripts can be seen clearly in the Zmap browser screenshot. We have also updated the figure legend to attempt to make navigation of the diagram clearer. However, all GENCODE lncRNAs sequences extended are publicly available to the community in major databases such as UCSC, Ensembl and HGNC.

5. Comparison of results of RACE-seq data generated by using pooled RNA with GTEx normal where individual samples are sequenced does not make any sense as the variation in transcript structure can be sample specific.

This is exactly the goal of our analysis. Since the variation in transcript structure is sample specific, and GTEx covers a large range of tissue samples from multiple individuals, one could hypothesize that most splice junctions are already captured in GTEx. Our results show that this is not the case. Even when profiling tissues that are already included in GTEx, thanks to the sensitivity of RACE-seq we are able to discover a plethora of novel transcript isoforms.

6. The authors have shown that some of the transcripts merged to form a single larger transcript. For example: locus OTTHUMG00000009351 extended and now shows coding potential. The authors did not do any validation of such claims.

Since the GENCODE manual annotation group deals with both annotation of protein coding genes and lncRNAs, we use the same criteria to assign coding potential of coding genes as changing the biotype of lncRNAs to coding based on new evidence. The assignment of protein coding genes is done following strict guidelines as published on the Havana website (ftp://ftp.sanger.ac.uk/pub/annotation/Old_stuff/havana_guidelines_April_2012.pdf) and using specific evidence. Therefore we use the same evidence criteria for changing OTTHUMG00000009351 to coding as we do for annotating any gene as coding in the GENCODE gene set. Analysing Kuster and Pandey proteomics data has revealed that 82% of protein coding loci in GENCODE do have peptide support.

7. Some nested primer results (6%) did not match with standard RACE primer results. Did the authors confirm the findings using other nested primers?

We believe this slight discrepancy can be simply attributed to normal sample variation, as each library was not sequenced very deeply (~500,000 reads/tissue). Owing to both this limited depth and the relatively low specificity of standard RACE, some DNA molecules, although present in the first RACE, might not result in sequenced reads at that stage. The nested step of the RACE may amplify (“rescue”) those rare molecules, and, due to its improved specificity, enable their detection in the sequencing output.

8. None of the figures are labelled properly. Even after asking the authors to label the figures, only the main figures were labelled. Supplementary figures were still left unlabeled. With no proper labelling it is very difficult to know which figure the authors are referring to in the text.

Initially we did not embed labels into the figure image files. However, we did resize and label all of them, following the Reviewer’s suggestions. The size of the figures was equalized to 2,000px in width and labeled by adding figure numbers to the top left corner of all figures, including the supplementary ones. It seems that the updated figures were not sent out to all Reviewers.

Also, there are no legends for the supplementary figures.

The legends for the supplementary figures appeared at the end of the supplementary text in the initial submission. We moved them to the end of the main text to avoid further confusion.

9. The formatting of the figures is also not done properly. Every figure has a different size making it very difficult to review. The authors are requested to properly format their manuscript before submission.

We did resize all figures, including the supplementary ones (see response to point #8). The size of all figures was equalized to 2,000 pixels wide, as requested by the Reviewer.

Reviewer #2 (Remarks to the Author):

Despite advances in transcriptomics, generating complete and accurate annotations of long noncoding RNAs (lncRNAs) is still difficult. Lagarde et al introduce RACE-seq, a novel method for targeted sequencing and annotation of otherwise difficult to study genes. RACE-seq is an extremely sensitive method for detecting and annotating transcripts and significantly outperforms other methods in the detection of lncRNA transcriptional start and end sites. Sensitive detection of transcript boundaries is the major advantage of RACE-seq, as other current methods cannot match it for the sensitive definition of transcript ends.

I found the study interesting and thought-provoking and the data and analysis generally sound. My only question is whether the study is "important" enough for Nature Comm.

Since this methodology is used to generate data that is integrated in the annotation of the GENCODE reference gene set for human and mouse, used by many consortia including ENCODE, and displayed as default in the two major browsers, Ensembl and UCSC, we think it will be of interest to a wide audience. This is demonstrated by over 1000 citations our previous GENCODE lncRNA paper (Derrien et al., Genome Res. 2012, doi: 10.1101/gr.132159.111) had since Oct 2012. To highlight better its potential impact, we have also included the following statement in the manuscript (see "Data availability"):

"All curated novel isoforms were incorporated into the human GENCODE set (version 22 onwards)."

In addition to the methodological interest of our work, and of its interest as a resource, our work contributes importantly to understand the biology of lncRNAs as a class. It is widely assumed that lncRNAs are shorter than protein coding genes, have less exons (with a characteristic pattern of enrichment for two-exon genes), and less alternative splice isoforms. This could suggest a specific pattern of post-transcriptional regulation, distinct to that of protein coding genes. Our work shows that these features are mostly a consequence of incomplete annotation and not an intrinsic property of lncRNAs. Actually, lncRNAs are as long, and have similar number of exons and splice isoforms as protein coding genes. We believe this is an important result.

Most of the text is clear and well written, however the first line of the abstract and the first half of the introduction need editing to improve the English.

We have tried to improve the abstract and introduction.

Major points to address

- The manuscript could use more detail about the RACE-seq method.

An important aspect of the method that is not clear from the paper is whether RACE reactions are performed individually or as multiplexes. Was each RACE primer used in a separate reaction (i.e.: did targeting of ~400 loci require 400 RACE reactions) or were the RACE reactions performed with a pool of primers?

We thank the referee for this comment, as this step was not clearly described in the supplementary text. This section has now been updated ("*RACE Reactions*" paragraph):

"RACE and nested RACE specific primers were synthesized by Life Technologies Europe BV and were diluted to a final concentration of 200 nM. All RACE reactions were performed in independent wells on 384 well plates as follows.

Double-stranded cDNA synthesis, adaptor ligations to the synthesized cDNA and 12.5µl final volume RACE reactions were performed according to the manufacturers' instructions. Nested RACEs were performed with 0.5µl of the initial RACEs in a final volume of 12.5µl. The cycling parameters were: RACE 5x(94°C 30" - 70°C 30" - 72°C 3'), 5x(94°C 30" - 68°C 30" - 72°C 3'), 20x(94°C 30" - 66°C 30" - 72°C 3'); nested RACE 25x(94°C 30" - 68°C 30" - 72°C 3'). We then pooled by tissue 2µl of all nested RACE reactions and pools were purified using Qiaquick PCR purification kit (Qiagen, CA, USA) to proceed to 454+ library preparation."

The reactions were carried out individually and pooled after RACE amplification. RACE reactions were performed in plates and preparations was made in liquid handling robots. This methodology can be applied to hundreds of genes simultaneously, as (1) nested RACE-Seq is both highly sensitive and specific, as reported in our manuscript, and (2) PCR reactions can be scaled up as they are done using liquid handling robots.

For others in the field considering adopting this technique an important aspect will be how easy/laborious is it and how much extra work is required as one scales up the number of gene loci targeted.

RACE amplifications were performed in plates and all the steps were done in a liquid handling robot, which allows easily to scale up the number or PCR reactions. The only step done manually is the transfer of 0.5 µl of the first RACE into the second race. In addition, it should be noted that the entire experimental design pipeline is automatized, as mentioned in the supplementary text.

- The authors may wish emphasise their results about the greatly increased number of lncRNA isoforms per locus. Previous findings, (i.e.: Derrien et al) that there are few lncRNA isoforms per loci and much less than from expressed from coding loci, may be an artifact.

Following the Reviewer’s suggestion, we have performed the required additional analysis and added the following text to the manuscript:

“Derrien et al. made the striking observation that lncRNAs have a very strong bias towards two-exon structures and exhibit less alternatively spliced isoforms per locus compared to protein-coding genes. Our results suggest that these are artifacts arising from inaccurate annotation of lncRNA transcript structures, since the biases towards both two-exon transcripts and isoform-poor genes disappear in the post-RACE-Seq transcripts (Figures 4c and 4d).”

A supporting figure (4d) has also been added to the paper:
(figure 4d)

- The comparison between RACE-seq and CaptureSeq clearly shows RACEseq is superior at detecting transcript ends, especially when they are previously un-annotated. However I am not convinced by the analysis suggesting RACE-seq is more sensitive at transcript detection in general, as I don't think this is an apples-to-apples comparison. The RACE-seq is targeting ~400 loci, the compared CaptureSeq design targeted every known lncRNA, >16000 loci and the

sequencing was not saturating. A CaptureSeq design targeting only ~400 loci may well have shown similar sensitivity to RACE-seq.

I realise the authors have attempted to correct for this, but I don't see how this has created comparable data.

We agree with the reviewer, and have added a clear disclaimer at the end of the corresponding section in the main text:

"It is important to stress that the isoform discovery rate of both methods is negatively correlated with the number of targeted genes (16,453 in Clark et al.'s study, 398 in the present one), owing to the limited sequencing depth they rely on. These differences are not fully accounted for in our analysis, and may therefore heavily favor our method over CaptureSeq in this comparison. "

Minor comments

- Regarding the description of CaptureSeq in the introduction. The probes hybridise to cDNA not RNA.

The manuscript has been corrected accordingly.

Results

- "are designed in-silico, and picked along the transcript sequences". In this sentence, do you mean "positioned" not "picked"?

The manuscript has been corrected accordingly.

- the results state that primers with >95% identity to other transcribed region were not used. However the supplementary methods give a value of >80%. Can the authors please check this and correct if need-be.

This issue has been corrected in the main text of the manuscript.

- The authors state "Seventy- five novel transcripts extended their parent locus in both 5' and 3' over their entire length". I don't think stating "over their entire length" makes sense and I'm not sure the authors need to say this.

The manuscript has been corrected accordingly.

- The results in Figure 4b suggest that annotation via RACE-seq leads to lncRNAs being slightly longer than before. It would be useful if the authors could confirm this increase statistically.

We thank the reviewer for pointing that out. We have amended the text of the manuscript accordingly:

"[...] and the median length of the transcripts slightly increased from 623 to 704, although not significantly ($p=0.7$, Wilcoxon rank sum test with continuity correction) (Figure 4b). It should be mentioned that RACE, by design, does not produce full-

length, TSS-to-TTS transcripts. This is because RACE products, by definition, start at their originating primer's position along the targeted transcript. Therefore, we speculate that the length of post-RACE transcripts is heavily underestimated."

- Figure 3a - please change the colours in the key to better represent those used in the figure. Figure 3a has been changed following the reviewer's suggestion:

(figure 3a)

- The supplementary word doc, still contains tracked comments from the first author, the authors may wish to remove this.

Tracked comments have been removed from the manuscript.

Reviewer #3 (Remarks to the Author):

A. Summary of the key results

The authors describe RACE-seq, which combines RACE with RNA sequencing to discover rare transcript isoforms and to accurately define the 5' and 3' ends of transcripts. Using this technique, they find that most targeted lncRNA loci are extended in the 5' and 3' direction, and discover more than 2,500 novel alternative transcripts.

B. Originality and interest: if not novel, please give references

The purpose of this work is highly interesting: lncRNA transcripts are poorly annotated, and in particular it is important to know the exact 5' end of the lncRNA, as it enables an analysis of regulatory control sequences in the proximal promoter region of the lncRNA.

However, the originality of the work is limited, as the following paper from 2009 already described the combination of 5' RACE with high-throughput sequencing:

Olivarius S, Plessy C, Carninci P: "High-throughput verification of transcriptional starting sites by Deep-RACE." *Biotechniques* 46(2): 130-132 (2009).

Carninci et al. only presented the 5'RACE analysis of 17 protein-coding genes in HepG2 cells, and compared single short-read Illumina sequencing against Sanger sequencing. They used modified primers with Illumina adapter sequences in order to directly generate sequencing libraries. Our method uses standard RACE and transcript specific inner primers, being agnostic of the long read sequencing platform. In addition, our submitted publication utilizes long read 3' and 5' RACE-Seq of lncRNA genes on a much larger scale (398 genes assayed in both RACE directions and in 8 different tissues). The main result of this is not only deep analysis of the lncRNAs, but also facilitates a full-length annotation of the transcripts thanks to the long read data.

A relevant section has been included on page 3 of the main text:

"The possibility of combining RACE with high-throughput sequencing was previously described by Carninci et al²³. However, this study presented only 5'RACE analysis of 17 protein coding genes and compared single short-read Illumina sequencing with Sanger sequencing, thus did not fully explore the high-throughput potential of this approach. In contrast, we applied RACE-Seq on a selection of 398 low-expression lncRNA loci from the reference GENCODE v7 catalog⁷ that lacked typical landmarks of Cap Analysis of Gene Expression (CAGE) and Gene Identification Signature-Paired End diTagging²⁴ (GIS-PET) tags supporting their 5' and 3' end, respectively."

C.Data & methodology: validity of approach, quality of data, quality of presentation

The quality of presentation could be improved. The English is sometimes rather awkward, and some of the figures are difficult to understand. In particular, it would be better to show Figure 2(a) as a Venn diagram, and to change the color legend of Figure 3(a). Also Figure 5(a) and (b) are hard to understand. Tables S4 and S5 should be shown in the main text, as the paragraph on the comparison to PolyA-Seq is quite difficult to understand in its current form.

We thank the reviewer for the suggestions regarding Figure 3a, which was modified according to the Reviewer's suggestions:

(figure 3a)

We also converted the bar plots shown in Figure 2a into Venn diagrams:

(figure 2a)

We have improved the readability of figures 5a and 5b:

(figure 5a)

(figure 5b)

Figures 5a and 5b could also be easily represented using UCSC Browser screenshots, if preferred.

Following the Reviewer's suggestion, tables S4 and S5 have been moved to the main text, as Figures 2c and 2d.

I was confused by the sentence "This is probably due to the addition to many new short alternative variants, many of them anchored at their originating RACE primer location." What does this refer to?

We have clarified this statement on page 6 of the manuscript:

"It should be mentioned that RACE, by design, does not produce full-length, TSS-to-TTS transcripts. This is because RACE products, by definition, start at their originating primer's position along the targeted transcript. Therefore, we speculate that the length of post-RACE transcripts is heavily underestimated."

The sentence "The function of this lncRNA ... and mediate the degradation of mitochondrial antiviral signals." does not add much to the theme of this manuscript, and can be dropped.

We agree with the reviewer, this sentence has been removed.

D. Appropriate use of statistics and treatment of uncertainties

No comments.

E. Conclusions: robustness, validity, reliability

I am surprised by the large fraction of off-target RACE amplification (though mitigated by the nested primer design). Is there any explanation for this?

We believe this is mainly due to the very low expression level of our targets. We are currently performing a preliminary nested RACE-Seq study on a set of 550 highly expressed protein-coding loci, using the exact same primer design pipeline parameters as in the present study. Consistent with this hypothesis, we observe >75% of reads on target (data not shown), compared to 36.4% in the present study.

Also I am concerned about the scalability of this approach. In spite of the high-throughput sequencing employed, only 398 lncRNA loci could be investigated.

Only 398 lncRNA were targeted because it was intended to be a pilot study and not because the methodology is limited to this number of loci. RACE amplifications were performed in plates and all the steps are done in a liquid handling robot, which allows easily to scale up the number of PCR reactions. Please refer to responses to reviewer #1's comments for more details.

F. Suggested improvements: experiments, data for possible revision

I can understand that RACE-Seq is successful at extending lncRNA loci at their 5' end if they are not supported by CAGE, but why is the same true for the 3' end?

ESTs, on which GENCODE v7 is mostly based, often arise from oligo-dT priming. Therefore, the 3' half of a transcript is likely to be better represented in EST databases than its 5'

counterpart. This has been widely reported in the genomics community. As a consequence of this 3' bias, a transcript model which is complete at its 5' end (i.e., CAGE-supported), is also likely to be complete at its 3' end. Under this assumption, CAGE supported transcripts are expected to be more challenging to extend further in both 5' and 3' directions, which is what we observe in our study. We clarified this in the manuscript (page 4):

“Surprisingly, we observed a similar phenomenon at the 3' end of targeted loci: the mean/median genomic length of 3' extensions amounted to -15/-526 and +225/+8,518 (positive values correspond to novel Transcription Termination Sites (TTSs) downstream of the annotated locus’), respectively for CAGE-supported and unsupported loci. We speculate that this observation is due to the pre-RACE-Seq GENCODE set being mostly based on oligo-dT-primed ESTs, which tend to cover preferentially the 3' end of transcripts. As a consequence of this bias, a transcript model that is complete at its 5' end (i.e., CAGE-supported), is also likely to be complete at its 3' end, which is consistent with our results.”

Regarding the comparison to PolyA-Seq: I would expect that many lncRNA transcripts are do not have a poly(A)-tail, which may explain the low coverage of the TTSs by PolyA-Seq.

The comparison of Figures 2c and 2d (figures S4 and S5 in the original submission) suggests that many of our targets *are* poly-adenylated, and that the PolyA-Seq datasets are probably not sequenced deep enough to detect them. We have clarified this statement on page 5 of the manuscript:

“This indicates that the low Merck PolyA-Seq coverage of our TTSs is probably due to the limited depth and tissue coverage of PolyA-Seq compared to our RACE-Seq data.”

G. References: appropriate credit to previous work?

The work by Olivarius et al. (see above) should be cited.

The literature references have been updated by adding this publication.

H. Clarity and context: lucidity of abstract/summary, appropriateness of abstract, introduction and conclusions

Other than the points raised above, the logical flow of the paper is good.

REVIEWERS' COMMENTS:

Reviewer #1 (Remarks to the Author):

In manuscript "Extension of human lncRNAs transcripts by RACE coupled with long read high-throughput sequencing (RACE-Seq)", the authors have selected 398 lowly expressed lncRNAs and performed RACE-seq to identify new transcript variants and the exact size of the lncRNAs using long read high throughput sequencing, i.e. 454. The authors have addressed most of the concerns raised in the initial submission but a few still remain.

Major Issues:

1. It is still unclear whether anyone would actually use this technique to identify proper lncRNA structure. The authors emphasize that this is a high throughput technique, however it still requires generating primers for each transcript (manually or automated), running PCR reactions separately for each transcript (both RACE and nested RACE) followed by sequencing. Definitely this technique is faster than regular RACE followed by sanger sequencing, however with rapid advances in the field of long read sequencing, this technique might not be useful.
2. It would be beneficial to the community if the authors host the sequencing results via a portal or even a supplementary table.
3. The authors have said they selected lowly expressed lncRNA for the analysis but some of the targets have expression of more than 200 RPKM.

Reviewer #2 (Remarks to the Author):

The authors have done a good job of addressing my queries and those of the other reviewers.

Rereading the manuscript I have a set of fairly minor corrections the authors should make prior to publication.

1. Abstract line 45 says: "novel spliced transcripts t, in contrast to current assumption" Please remove "t"
2. The introduction is still a bit clunky. There are unnecessary commas, a few missing words and some unnecessary words. Please have a native English speaker help edit it.
A couple of examples:
i.e. lines 84-85: "transcript enrichment by the hybridization of the cDNA" - could be "transcript enrichment by the hybridization of cDNA"
i.e. lines 89-90: "Determining such ends, is essential to fully..." No need for a comma here.

3. Line 97: Reference to Olivarius paper (ref23), should be "Olivarius et al", not Carninci et al

4. In their response to my first major query the authors wrote:

"The reactions were carried out individually.....done using liquid handling robots."

Much of this paragraph is a better explanation of how the RACE is done and can be scaled that exists in their manuscript. I would encourage the authors to add a slightly modified version of the paragraph to the manuscript.

5. Discussion Line 350: This sentence doesn't make sense, please fix.

6. References Line 510: The authors should change their nanopore sequencing reference to a more recent paper that better represents the current capabilities of Nanopore sequencing. I suggest Bolisetty et al in Gen Biol: <https://genomebiology.biomedcentral.com/articles/10.1186/s13059-015-0777-z>

Methods supplement:

7. Lines 6-8: The authors state they filtered for lncRNAs with an RPKM above 5 and then picked the top lncRNAs ranked by expression. However the manuscript states these were lowly expression lncRNAs? Did the authors mean an RPKM of under 5?

8. Lines 48-49: "All RACE reactions were performed in independent wells on 384 well plates as follows."

I think it would be better to state "Each RACE reaction was performed in an independent well on a 384 well plate as follows". This more clearly states that each RACE reaction was performed separately.

Reviewer #3 (Remarks to the Author):

The authors have adequately addressed the issues that I raised in my review. However, the novelty of the proposed method remains limited. In their response, the authors write that Olivarius et al. only presented 17 protein-coding genes. But if the current work is intended as a pilot study (as the authors write), then the number of genes is not so relevant. Also, the fact that the proposed method is agnostic of the sequencing platform seems of limited importance. I can accept the manuscript in its current form, but a more specialized journal than Nature Communications seems more appropriate.

Minor comment: Please refer to the 2009 Biotechniques paper as Olivarius et al. rather than Carninci et al., as Olivarius is the first author.

Reviewers' comments:
(Authors' responses are in green)

Reviewer #1 (Remarks to the Author):

In manuscript "Extension of human lncRNAs transcripts by RACE coupled with long read high-throughput sequencing (RACE-Seq)", the authors have selected 398 lowly expressed lncRNAs and performed RACE-seq to identify new transcript variants and the exact size of the lncRNAs using long read high throughput sequencing, i.e. 454. The authors have addressed most of the concerns raised in the initial submission but a few still remain.

Major Issues:

1. It is still unclear whether anyone would actually use this technique to identify proper lncRNA structure. The authors emphasize that this is a high throughput technique, however it still requires generating primers for each transcript (manually or automated), running PCR reactions separately for each transcript (both RACE and nested RACE) followed by sequencing. Definitely this technique is faster than regular RACE followed by sanger sequencing, however with rapid advances in the field of long read sequencing, this technique might not be useful.

To the contrary, we expect the combination of RACE-Seq with longer read sequencing to make our method all the more pertinent. RACE-Seq enables the detection of very rare transcripts, and longer reads would avoid the need to assemble reads into full-length transcript structures.

2. It would be beneficial to the community if the authors host the sequencing results via a portal or even a supplementary table.

All sequencing files have been uploaded to the European Nucleotide Archive under accession ERP012249, and are available for download at the following URL: <http://www.ebi.ac.uk/ena/data/view/ERP012249>

We are in the process of creating a simple web portal (URL: http://public-docs.crg.es/rquigo/Papers/2016_lagarde-uszczynska_RACE-Seq/) to facilitate users' exploration of this dataset.

3. The authors have said they selected lowly expressed lncRNA for the analysis but some of the targets have expression of more than 200 RPKM.

As stated in the Methods section, under "Target selection and primer design", the selection of targets was not based on their low expression level, but rather on the absence of ENCODE CAGE tags in their annotated TSS's vicinity. It so happened that most of them exhibited low RPKMs in HBM. The distribution of RPKMs for the 398 targets (averaged over matched tissues in the HBM dataset), as computed from supplementary table 1, is summarized below:

Min.	1st Qu.	Median	Mean	3rd Qu.	Max.
2.856	4.563	8.301	27.36	16.5	1,981

Reviewer #2 (Remarks to the Author):

The authors have done a good job of addressing my queries and those of the other reviewers.

Rereading the manuscript I have a set of fairly minor corrections the authors should make prior to publication.

1. Abstract line 45 says: "novel spliced transcripts t, in contrast to current assumption" Please remove "t"

This has been corrected.

2. The introduction is still a bit clunky. There are unnecessary commas, a few missing words and some unnecessary words. Please have a native English speaker help edit it.

A couple of examples:

i.e. lines 84-85: "transcript enrichment by the hybridization of the cDNA" - could be "transcript enrichment by the hybridization of cDNA"

i.e. lines 89-90: "Determining such ends, is essential to fully..." No need for a comma here.

We have rewritten most of the introduction with the help of a native English speaker, and hope to have improved it.

3. Line 97: Reference to Olivarius paper (ref23), should be "Olivarius et al", not Carninci et al

This has been corrected.

4. In their response to my first major query the authors wrote:

"The reactions were carried out individually.....done using liquid handling robots."

Much of this paragraph is a better explanation of how the RACE is done and can be scaled that exists in their manuscript. I would encourage the authors to add a slightly modified version of the paragraph to the manuscript.

We have improved the Methods section accordingly.

5. Discussion Line 350: This sentence doesn't make sense, please fix.

This sentence has been edited accordingly.

6. References Line 510: The authors should change their nanopore sequencing reference to a more recent paper that better represents the current capabilities of Nanopore sequencing. I suggest Bolisetty et al in Gen Biol: <https://genomebiology.biomedcentral.com/articles/10.1186/s13059-015-0777-z>

This has been corrected.

Methods supplement:

7. Lines 6-8: The authors state they filtered for lncRNAs with an RPKM above 5 and then picked the top lncRNAs ranked by expression. However the manuscript states these were lowly expression lncRNAs? Did the authors mean an RPKM of under 5?

As stated in the Methods section, under "Target selection and primer design", the selection of targets was not based on their low expression level (we did select those with an RPKM *above* 5, though), but rather on the absence of ENCODE CAGE tags in their annotated TSS's vicinity. It so happened that most of them exhibited low RPKMs in HBM. The distribution of RPKMs for the 398 targets (averaged over matched tissues in the HBM dataset), as computed from supplementary table 1, is summarized below:

Min.	1st Qu.	Median	Mean	3rd Qu.	Max.
2.856	4.563	8.301	27.36	16.5	1,981

8. Lines 48-49: "All RACE reactions were performed in independent wells on 384 well plates as follows."

I think it would be better to state "Each RACE reaction was performed in an independent well on a 384 well plate as follows". This more clearly states that each RACE reaction was performed separately.

The text has been edited accordingly

Reviewer #3 (Remarks to the Author):

The authors have adequately addressed the issues that I raised in my review. However, the novelty of the proposed method remains limited. In their response, the authors write that Olivarius et al. only presented 17 protein-coding genes. But if the current work is intended as a pilot study (as the authors write), then the number of genes is not so relevant. Also, the fact that the proposed method is agnostic of the sequencing platform seems of limited importance. I can accept the manuscript in its current form, but a more specialized journal than Nature Communications seems more appropriate.

Minor comment: Please refer to the 2009 Biotechniques paper as Olivarius et al. rather than Carninci et al., as Olivarius is the first author.

The text has been edited accordingly